# Molecular Testing in CML between Old and New Methods: Are We at a Turning Point?

**DOI:** 10.3390/jcm9123865

**Published:** 2020-11-27

**Authors:** Simona Soverini, Simona Bernardi, Sara Galimberti

**Affiliations:** 1Department of Experimental, Diagnostic and Specialty Medicine, Hematology/Oncology “Lorenzo e Ariosto Seràgnoli”, University of Bologna, 40138 Bologna, Italy; simona.soverini@unibo.it; 2Department of Clinical and Experimental Sciences, University of Brescia, Bone Marrow Transplant Unit, ASST Spedali Civili, 25123 Brescia, Italy; 3Centro di Ricerca Emato-Oncologica AIL (CREA), ASST Spedali Civili, 25123 Brescia, Italy; 4Department of Clinical and Experimental Medicine, Hematology Unit, University of Pisa, 56126 Pisa, Italy; sara.galimberti@unipi.it

**Keywords:** chronic myeloid leukemia, tyrosine kinase inhibitors, digital PCR, next-generation sequencing

## Abstract

Molecular monitoring of minimal residual disease (MRD) and *BCR-ABL1* kinase domain (KD) mutation testing have a well consolidated role in the routine management of chronic myeloid leukemia (CML) patients, as they provide precious information for therapeutic decision-making. Molecular response levels are used to define whether a patient has an “optimal”, “warning”, or “failure” response to tyrosine kinase inhibitor (TKI) therapy. Mutation status may be useful to decide whether TKI therapy should be changed and which alternative TKI (or TKIs) are most likely to be effective. Real-time quantitative polymerase chain reaction (RQ-qPCR) and Sanger sequencing are currently the gold standard for molecular response monitoring and mutation testing, respectively. However, in recent years, novel technologies such as digital PCR (dPCR) and next-generation sequencing (NGS) have been evaluated. Here, we critically describe the main features of these old and novel technologies, provide an overview of the recently published studies assessing the potential clinical value of dPCR and NGS, and discuss how the state of the art might evolve in the next years.

## 1. Introduction

Chronic myeloid leukemia (CML) is regarded as a paradigm of precision medicine. The hallmark of the disease, the *BCR-ABL1* fusion gene, resulting from the t(9;22) chromosomal translocation and encoding a deregulated tyrosine kinase responsible for leukemic transformation [1], provides both an ideal therapeutic target and a robust marker of minimal residual disease (MRD). Three generations of tyrosine kinase inhibitors (TKIs) are now available for the treatment of CML patients [2]. Thanks to their remarkable efficacy, most CML patients face a near-normal life expectancy and a non-negligible proportion of them may even discontinue the treatment, achieving so-called “treatment-free remission” (TFR) [2].

Optimization of CML treatment throughout the years has been made possible by the forward-looking implementation and standardization of molecular monitoring strategies. Since the International Randomized study of Interferon-alpha plus cytarabine (IFN-alpha plus ara-C) versus STI571 (imatinib mesylate) (IRIS) study, which led to the approval of the first TKI, imatinib [3], real-time quantitative reverse transcription polymerase chain reaction (RT-qPCR)—a newborn technology at those times—has been used to monitor patient response in terms of reduction of *BCR-ABL1* transcript levels (molecular response (MR)). It was soon realized that MR measurement by RT-qPCR could best stratify patients in terms of “depth” of response as compared to the classical hematologic and cytogenetic (reduction of the percentage of bone marrow metaphases in which the t(9;22) translocation was detectable) assessments. Over the years, time-dependent MR milestones have been defined that harbor prognostic significance, draw the ideal trajectory toward optimal response, and lay the foundations for TFR [2,4]. They include major molecular response (MMR or MR**^3^**; defined as a 3 log reduction of *BCR-ABL1* transcript levels from the standardized baseline defined by the International Scale (IS), that is, *BCR-ABL1* < 0.1%) and deep molecular response (DMR), further stratified into MR**^4^** (4 log-reduction or *BCR-ABL1* < 0.01%), MR^4.5^ (4.5 log-reduction or *BCR-ABL1* < 0.032%), MR5 (5 log-reduction or *BCR-ABL1* < 0.001%). Since the early 2000s, intensive international cooperative efforts have been devoted to the technical optimization of the assays and to the standardization of MR definitions and acceptability criteria, so as to maximize precision, accuracy, and interlaboratory reproducibility of results.

Molecular testing is important also in the setting of patients who do not achieve an optimal response to TKI therapy. Such patients may have acquired point mutations in the kinase domain (KD) of *BCR-ABL1* that impair TKI binding. Detailed lists of mutations conferring resistance to each TKI are currently available [5]. Although other mechanisms of resistance have been reported, *BCR-ABL1* KD mutations are, at present, the only “actionable” one, since the detection of a TKI-resistant mutation mandates a change of therapy and the type of mutation may guide the selection of the second- or subsequent-line TKI [6]. Sanger sequencing has long been considered the gold standard for *BCR-ABL1* KD mutation screening. However, this sequencing technique has a relatively poor sensitivity. Moreover, in cases who have failed multiple lines of therapy and may harbor two or more mutations, Sanger sequencing must often be preceded by a cumbersome process of cloning in order to assess whether mutations are on the same (“compound”) or on different (“polyclonal”) BCR-ABL1 molecules. Assessment of clonal configuration is important, because compound mutations have been shown to be highly resistant to first-, second-, and in some cases even third-generation TKIs [7].

The flow chart of routine assessments in CML is reported in Figure 1.

Technologies for MR monitoring and mutation testing have, in the meantime, evolved. Here, we describe some of the novel technologies that are going to impact most the way we measure response to therapy and identify TKI-resistant mutations, and we critically review the results of the studies that have already been performed to evaluate their application in CML.

## 2. MR Monitoring

In the last few years, many studies have been conducted in order to overcome the intrinsic limitations of “classical” quantitative PCR (RT-qPCR) [8,9] (such as the limit of detection fixed at three copies of *BCR-ABL1* transcript, the need for a standard curve, or the sensitivity to inhibitors), both improving its performance [10,11] and looking at further novel technologies [12]. In this context, digital PCR (dPCR) seems to be one of the most promising tools [13,14]. Indeed, dPCR is considered the third generation of PCR. It was developed to overcome certain limits of conventional amplification techniques, particularly to allow detection of small amounts of target nucleic acids [15]. Quantification by dPCR is based on the fact that the random distribution of molecules in many partitions is regulated by the Poisson’s distribution. The number of partitions and the partitioning strategy may vary according to the platforms. The first-generation dPCR platforms presented a moderate number of partitions (<200) and were based on a physical separation using chips with microchannels (microfluidics-based dPCR). The advent of the second-generation dPCR platforms increased the number of the micro-reactions (up to 20,000) and improved the partitioning strategies: either a physical separation of the different partitions using chips with micro-chambers (chip-based dPCR) or a separation using automated partitioning by creating a “water-in-oil” emulsion (droplet-based dPCR). Recently, the third-generation dPCR platforms were developed, and the most important innovations are the increased number of partitions, up to 10^6^, with a low thermal mass and the possibility to perform a fully integrated process within a single instrument that combines consumable sample loading, thermal cycling image acquisition, and analysis software. This innovation creates a streamlined workflow that reduces contaminations and human handling errors and has a unique capability to decrease the time to result [16].

Each partition acts as a single PCR micro-reaction chamber, and partitions containing the amplified target are then identified by fluorescence emission. The PCR-positive partition number determines the absolute quantity of target without a need for external calibration [17]. The sensitivity of dPCR has been reported to be comparable to or sometimes higher than that of RT-qPCR [18,19]. In acute lymphoblastic leukemia and non-Hodgkin’s lymphoma, for example, dPCR was able to recover about one-quarter of cases that were scored as “positive non-quantifiable” (a definition that applies to borderline positive/negative samples) by conventional RT-qPCR, demonstrating that dPCR has a sensitivity about 1.5 log higher than that of RT-qPCR [20,21].

In recent years, different groups approached dPCR for quantification of the *BCR-ABL1* transcript in CML patients. In the Imatinib Suspension And Validation (ISAV) study, which enrolled subjects on imatinib therapy for more than 2 years and with undetectable *BCR-ABL1* by RT-qPCR for at least 18 months, Mori et al. demonstrated that a more successful selection of candidates for TKI discontinuation (TFR) was offered by the application of the Fluidigm dPCR platform. This platform is a first-generation dPCR relying on a relatively small number of multiple parallel micro-reactions. Overall, the large majority of cases which remained in stable undetectable DMR [22] were dPCR-negative, with a higher number of patients losing TFR in the dPCR-positive than in the dPCR-negative group (68% vs. 43%) [23]. Moreover, dPCR status resulted in a difference of approximately 10–20% between subjects maintaining and losing TFR, with all subjects < 45 years and dPCR-positive who had to restart TKI therapy by 15 months [23].

The application of the Fluidigm platform was also experienced by other groups, confirming the good accuracy and sensitivity of dPCR in the setting of *BCR-ABL1* transcript measurement. In the ENEST next trial, which evaluated the rate of DMR in patients treated with nilotinib as first-line therapy, dPCR detected residual *BCR-ABL1* transcripts in 39.4% of samples that were estimated to be in confirmed MR^4.5^ [22] by RT-qPCR, thus demonstrating the feasibility of monitoring very low *BCR-ABL1* transcript levels using dPCR [24].

Later, Goh and colleagues confirmed the capability of nanofluidic dPCR to monitor the continuous decline in MRD levels even after they became undetectable by conventional RT-qPCR. Indeed, when 62 samples were screened by RT-qPCR and dPCR, the latter, preceded by a pre-amplification step, showed a 2–3 log improvement in sensitivity, with 75% of samples defined as negative by RT-qPCR that showed detectable *BCR-ABL1* transcripts by dPCR [25]. In another study that compared RT-qPCR to dPCR, the correlation between the two methods reached 99%, but only dPCR was able to successfully predict loss of MR^4^, showing the increase in *BCR-ABL1* transcripts 3 months earlier than RT-qPCR in 4/10 cases [26].

The application of dPCR also improved MRD monitoring in pediatric CML cases. An interesting approach demonstrated the feasibility of using dPCR for the amplification of relatively long amplicons, thus allowing to monitor *BCR-ABL1* not on complementary DNA (cDNA) but on genomic DNA (gDNA), which might be relevant if we consider that CML leukemic stem cells hidden in the hypoxic bone marrow niche are unable to synthesize *BCR-ABL1* messenger RNA (mRNA) and the corresponding fusion protein [27]. Differently from adults, in pediatric patients, the *ABL1* and *BCR* breakpoint cluster regions are both positioned in genomic regions rich in Alu sequences, which might impair the perfect match of primers and the accurate quantitation of *BCR-ABL1* transcript [28]. Of 687 tested samples, 47 were negative both on gDNA and on cDNA, and 1% of cases were quantifiable by RT-qPCR, but not using the gDNA-based assay. Conversely, when gDNA-based PCR was used, it was able to quantify 9% of samples that tested negative at the cDNA level, thus demonstrating that the most sensitive approach might be the combination of PCR on both cDNA and gDNA [29].

The possibility of detecting *BCR-ABL1* on gDNA rather than on cDNA is also very interesting in adults, because it was previously reported that patients in long-term TFR who do not show detectable *BCR-ABL1* transcripts on cDNA can be positive on gDNA, especially if the latter is extracted from the lymphoid compartment where B cells are more often *BCR-ABL1*-positive than T cells [29]. A gDNA-based approach was used also to demonstrate the increase in sensitivity enabled by dPCR as compared to fluorescence in situ hybridization (FISH) for reliable quantification of the major breakpoint cluster region and for the detection of the *BCR-ABL1* fusion gene [30,31].

The application of a chip-based dPCR platform for the detection of *BCR-ABL1* transcripts was recently set up by our group. Bernardi and colleagues underlined the capability of dPCR to offer a precise, sensitive, and accurate quantification of *BCR-ABL1* transcripts in both peripheral blood cells and circulating exosomes [32,33] (Figure 2).

Indeed, it was clearly demonstrated that CML cells may release extracellular vesicles that affect both in vitro and in vivo tumor progression [34]. The number of circulating exosomes is significantly higher in CML patients at diagnosis (2685.54 copies/mL plasma) as compared to patients in early phases of treatment (1969.605 copies/mL plasma) or in DMR (286.695 copies/mL plasma) and as compared to healthy individuals (median, 172.44 copies/mL plasma). In 10 CML cases, *BCR-ABL1* transcript was measured both on whole peripheral blood and on exosomes; in three cases, dPCR on exosomes was unable to detect any *BCR-ABL1* transcript, while another sample resulted positive only on vesicles. In the remaining cases, both compartments were positive, even if with a wide range of median values, thus suggesting that measure of the target in both compartments would probably be more informative [32].

Moreover, in another study, Bernardi et al. showed for the first time how to improve the selection of CML patients eligible for TFR by using a second-generation dPCR platform. While RT-qPCR failed to predict MR^3^ loss after TKI discontinuation, dPCR showed that 48% of patients with *BCR-ABL1* values ≥ 0.468 but only 14% of those with *BCR-ABL1* values < 0.468 failed to maintain TFR [35,36]. This was later confirmed also by the French cooperative group, who explored the application of dPCR in the prediction of the achievement of TFR in patients undergoing imatinib therapy discontinuation. They indeed showed that the duration of TKI therapy (≥74.8 months) and *BCR-ABL1* values measured by dPCR were the only two significantly predictive factors of molecular recurrence [37].

dPCR also proved its value when patients in DMR were analyzed in parallel using RT-qPCR and dPCR for the monitoring of *BCR-ABL1* transcripts and by flow cytometry for the quantitation of circulating residual leukemia stem cells (LSCs) identified by the CD26 surface marker. The authors observed that both dPCR and LSC quantitation by flow cytometry were more sensitive than RT-qPCR in identifying patients with residual disease, scoring as positive a series of samples where the *BCR-ABL1* transcript was undetectable by RT-qPCR [38].

Very recently, a third-generation dPCR chip-based platform was also tested for MRD monitoring in CML. Indeed, the novel microfluidic array partitioning device was capable of precisely quantifying *BCR-ABL1* transcripts down to a 0.01% abundance, with high reproducibility across multiple replicates [39].

In addition to the technical limitation in the detection of very low levels of *BCR-ABL1* transcripts, another pitfall of RT-qPCR seems to lie in the different efficiency of amplification of the e14a2 transcript variant as compared to the shorter e13a2—a problem that might negatively impact the clinical management of CML patients [40]. This hypothesis was investigated by Kjaer et al., who reported this discrepancy when using RT-qPCR and demonstrated the capability of dPCR to overcome this limitation [41]. These results were also confirmed by another study where our group showed that no differences in terms of quantification were found by dPCR between cohorts with the e13a2 versus the e14a2 transcript, in contrast to what happened when RT-qPCR was used [42].

Considering these encouraging results, it is not surprising that new commercial assays have become available for *BCR-ABL1* transcript detection by dPCR. The QXDx BCR-ABL %IS by Bio-Rad is the first dPCR-based in vitro diagnostics product receiving the United States (US) Food and Drug Administration (FDA) clearance and the European Conformity (CE-IVD) mark. It is suitable for droplet-based platforms and its performance was very recently independently evaluated and reported [43]. The QXDx BCR-ABL %IS was compared with the gold standard RT-qPCR, and the two assays demonstrated a very strong correlation (*r* = 0.996) when quantifying *BCR-ABL1* transcript levels ranging from 20% to 0.002%. These results further support the introduction of dPCR as the standard of care for CML patient monitoring. In addition, some panels of experts recently underlined that coordinated efforts should be made for optimization and standardization of CML MRD monitoring by dPCR [44,45].

At the last American Society of Hematology (ASH) congress held in 2019, a discrepancy in the “molecular class” positioning when dPCR was used in comparison to RT-qPCR was reported (*p* < 0.0001). Indeed, 14.3% of samples already defined to be in MR^3.0^ by RT-qPCR appeared to have higher levels of *BCR-ABL1* transcripts by dPCR. On the other hand, 20.9% of cases in MR^4.0^ by RT-qPCR were not found to be in DMR by dPCR. This has a relevant clinical impact, considering that only patients who are in MR^4^ or better can be candidates for TFR [46].

At the 2020 meeting of the European Society of Hematology (EHA), a multicenter international study confirmed that dPCR is a valid alternative to RT-qPCR, demonstrating a low interlaboratory variation and a high assay linearity. Indeed, dPCR showed a detection rate of 90.9% at MR^4.5^, 81.2% at MR^4.7^ (corresponding to a 0.002% *BCR-ABL1/ABL1* level) [47], and 81% at MR^5.0^, compared to 90.9% at MR^4.5^, 90.9% at MR^4.7^, and 72.7% at MR^5.0^ of RT-qPCR (*p* < 0.05). Interestingly, the interlaboratory coefficient of variability of dPCR was lower than that measured for RT-qPCR (23.6% vs. 44.4% down to MR^4.0^) (*p* < 0.05) [48].

These data are constantly confirmed by new evidence and altogether stress the utility to move to dPCR for the monitoring of MRD in CML patients, particularly in subjects presenting a low level of *BCR-ABL1* transcript and potentially eligible for stopping TKI therapy. In fact, the biased diagnostic performance of the *BCR-ABL1* molecular detection and quantification may impact the inclusion of CML patients in therapy-stopping trials [49,50], which is currently one of the pivotal goals of CML management [4,51]. A comparison between the main features of RT-qPCR and dPCR is presented in Table 1.

## 3. BCR-ABL1 Mutation Testing

As mentioned above, *BCR-ABL1* KD mutation testing provides important information whenever therapeutic reassessment has to be considered. In this context, both next-generation sequencing (NGS) and dPCR have been explored as alternatives to Sanger sequencing.

When benchtop next-generation sequencers for diagnostic use first landed on the market, BCR-ABL1 KD mutation screening was one of the first applications to be explored in hematology given that mutation status was already recognized at those times as an essential piece of information in the clinical decision algorithms for TKI-resistant patients [6]. The power of NGS lies in the massively parallel sequencing of DNA molecules, including PCR amplicons, after they have been physically separated in space and clonally amplified. NGS of amplicons spanning the KD has the potential to pick any nucleotide substitution in the *BCR-ABL1* transcript with a sensitivity that can theoretically achieve 1%, to follow the kinetics of mutant transcripts over time and in relation to therapy, and to discriminate between compound and polyclonal mutations in cases of multiple substitution falling on the same sequence reads.

Between 2013 and 2016, a series of retrospective studies were published that aimed to investigate the use of NGS in patients who had developed resistance to TKI therapy. In particular, these studies engaged in the reanalysis by NGS of serial samples from patients who had acquired TKI-resistant mutations and had already been studied by Sanger sequencing. They demonstrated that NGS holds the potential to identify emerging mutations up to several months earlier than Sanger sequencing [52,53]. NGS was also capable of highlighting the presence of a TKI-resistant mutation in patients with a “warning” response who would ultimately fail TKI therapy [54]. Moreover, in the setting of failures, the greater sensitivity of NGS proved capable of providing more accurate information about mutation status both in Sanger-negative and in Sanger-positive cases, showing that, in the latter, mutations detectable by Sanger sequencing may be just “the tip of the iceberg” [55,56]. These retrospective data laid the foundation for two big prospective studies that were just recently published. In the first one [57], the King’s College group assessed 121 chronic-phase CML patients on first-line TKI therapy (mostly receiving imatinib) who were routinely screened for mutations by NGS regardless of their response. Mutations detectable by NGS were found to be associated with lower cumulative incidence of complete cytogenetic response, major molecular response, and shorter event-free survival (EFS) and progression-free survival (PFS). NGS proved capable to detect emerging mutations as early as after 3 months of therapy, and this was predictive of high risk of transformation; all patients indeed subsequently progressed to advanced phase disease after mutation detection. The second prospective study, a cooperative effort of 39 Italian groups [58], was aimed at testing the feasibility of routine NGS-based *BCR-ABL1* KD mutation screening within a lab network and to compare Sanger sequencing and NGS performance in a consecutive series of 236 CML patients eligible for mutation screening because of failure or warning response according to the 2013 ELN recommendations [59]. This study showed the accuracy and interlaboratory reproducibility in terms of mutation identification and quantitation down to a sensitivity of 3% and confirmed the advantages of NGS over Sanger sequencing. NGS detected low burden mutations in 22% of cases with failure or warning who were negative by Sanger sequencing. Additionally, almost half of the patients positive for mutations by Sanger sequencing had additional low burden mutations detectable by NGS only, and, in one-third of such cases, one of these mutations included a T315I or another mutation that, on the basis of its resistance profile, might influence the selection of the subsequent TKI. The availability of follow-up samples and clinical information enabled us to demonstrate that known TKI-resistant mutations remain consistently detectable and increase in burden whenever the TKI is not changed or is changed to another one not active against those mutations, thus underlining the clinical relevance of low-level mutations (at least of those present in 3% or greater of the *BCR-ABL1* transcripts).

Taken together, all the data summarized above have already resulted in NGS being included in position papers on *BCR-ABL1* mutation testing [5,60], as well as in the 2020 ELN recommendations [2].

Nevertheless, efforts aimed at enhancing our mutation-testing capabilities continue. On one hand, they have been and are being directed toward more sensitive and accurate NGS-based screening. Strategies of error-corrected sequencing have been described [61]. Moreover, third-generation sequencing on the MinION system, which exploits the use of nanopores (individual DNA molecules are scanned in real time as they translocate through a nanopore, forced by an electric field, and the sequence is determined by the extent the current is altered as each nucleotide passes through), was recently explored [62]. The MinION system is a small (it has the same size of a USB stick) and cheap device that enables fast and cost-effective (estimated cost per sample, 40 USD) sequencing of very long stretches of DNA (average read length, 1.7 kb), thus making the identification of compound mutations even more straightforward. The greatest limitation of nanopore sequencing in general is currently represented by the error rate, higher than that of NGS. This makes nanopore sequencing, at present, more suitable for the characterization of fusion transcripts or genomic breakpoints [31] than the identification of point mutations; however, companies are working hard to overcome this issue.

On the other hand, dPCR has been considered as a complementary or even alternative method. In the hematological scenario, dPCR has already been employed for detection of hotspot mutations in many genes, such as *JAK2* [19], *IDH1/IDH2* [63,64], *MYD88* [65], *B-RAF* [18,66], and *c-KIT* [67]. Each dPCR assay enables the detection of a given mutation by using specific primers and probes. However, this limitation, which results in the possibility to interrogate a limited number of key sequence positions, is counterbalanced by the advantages of dPCR over NGS. While the latter is more expensive and time-consuming and requires specific skills for both library preparation and data analysis (which limits NGS use to medium- to large-scale reference laboratories), dPCR is cheap and simple and guarantees a much shorter turnaround time. As anticipated above, each first- (imatinib) and second-generation (dasatinib, nilotinib, bosutinib) TKI exhibits a small, specific spectrum of resistant ABL1 mutations, and none is active against the “gatekeeper” T315I mutation, which can only be overcome by the third-generation TKI ponatinib [4]. Thus, the identification of this key ABL1 mutation has relevant clinical implications, because it must lead to an immediate switch from the ongoing TKI to ponatinib [2]. Small clones with the T315I mutation have recently been backtracked by using dPCR in patients with Philadelphia chromosome-positive acute lymphoblastic leukemia who received allogeneic stem-cell transplantation following dasatinib plus intensive chemotherapy. In this series of patients, the cumulative incidences of molecular and hematological relapse at 3 years were 60.3% and 25.1%, respectively [68]. All patients with the T315I mutation at molecular relapse rapidly experienced hematological relapse, whereas only two of the 10 patients without the T315I mutation had hematological relapse. This study suggests that dPCR can also be an effective and sensitive method for the detection of the T315I mutation in the setting of CML [68] (Figure 3).

At the 2019 ASH meeting, our group compared Sanger sequencing, NGS, and dPCR in CML patients with failure or warning responses to TKI therapy. The dPCR strategy was implemented using a single-tube assay multiplexing the detection and quantitation of the mutations conferring resistance to one or more second-generation TKIs: T315I/A, F317L/V/I/C, Y253H, E255K, F359V/I/C, E255V, and V299L. The assay was designed so that mutations clustered in five spatially distinct areas of the two-dimensional plot depending on their resistance profile (pan-resistant (T315I), dasatinib-resistant, nilotinib-resistant, dasatinib- and bosutinib-resistant, nilotinib- and bosutinib-resistant), in order to provide an indication as to which TKI (or TKIs) should be avoided, if any. The study confirmed the advantage of NGS over Sanger sequencing; however, it also highlighted that, in the setting of failures, low-level mutations detectable by NGS could have impacted the choice of the subsequent TKI in 9% of the patients. In samples positive for second-generation TKI-resistant mutations, a very good concordance was observed between dPCR and NGS, irrespective of type of mutation and variant allele frequency, as well as when reactions started from very low cDNA amounts (as little as 30 ng). In patients with a warning response, the aim of mutation testing is to identify those who need a change in therapy rather than a “watch and wait” approach, since detection of a TKI-resistant mutation is an indication for therapeutic switch. Bearing in mind that more than 50 different mutations may confer resistance to imatinib, NGS remains the best approach to highlight emerging mutations that are going to lead to treatment failure in patients with a warning response to imatinib. In contrast, in patients with a warning response to second-generation TKIs, dPCR might represent an attractive alternative. However, patients who would benefit most from dPCR are those who fail TKI therapy. Here, mutation testing aims to a timely and rational TKI switch (thus, turnaround time is critical) and only a limited spectrum of mutations impacts the selection of the second- or subsequent-line TKI [69].

At the same meeting, Vannuffel et al. [70] presented an innovative approach, called “drop-phase”, where droplet-based dPCR was used to identify compound mutations. If the two mutations whose clonal configuration must be established are known, “drop-phase” can be implemented performing a duplex reaction with dual-color probes specific for these two mutations. Presence of the mutations on the same DNA molecule results in more double-positive droplets than would be expected from the random colocalization in the same droplet of unlinked variants. “Drop-phase” is theoretically fast and easy to perform; however, probes for each and every mutation need to be synthesized, and they need to be labeled with nonoverlapping fluorophores when used in tandem. Considering all the possible mutation combinations, this may not be considered the most convenient strategy for the detection of compound mutations.

At the 2020 EHA meeting, we further confirmed the value of dPCR as a fast and effective technique for detecting the T315I mutation in another series of 44 CML samples. The sensitivity of dPCR was 0.02%, and overall dPCR allowed us to identify the T315I mutation in 16% of samples that were defined as wildtype by other methods. Indeed, when dPCR and Sanger sequencing results were compared, 25 samples showed concordant results, five were mutated by dPCR but not by Sanger, and no samples were wildtype by dPCR but mutated by Sanger. When dPCR was compared to NGS, 19 samples were concordant, two cases were mutated by NGS but wildtype by dPCR, and two additional cases were wildtype by NGS but mutated by dPCR. The variant allele frequencies (VAFs) in these two cases were 0.43% and 0.39%—values well below the detection limit of NGS (3%). Interestingly, out of the two cases where dPCR failed to detect the T315I on cDNA, one was found positive when gDNA was analyzed. Indeed, when applying dPCR to gDNA instead of cDNA, in 2/16 samples, the T315I mutation was detected only on gDNA (in both cases, the VAF was very low—0.01% and 0.05%). Thus, our experience demonstrated the value of dPCR in detecting the T315I mutation, with a sensitivity and accuracy not inferior to and sometimes better than those offered by NGS [71].

In summary, while the lower limit of detection of Sanger sequencing is routinely 20%, that of NGS is between 1% and 3%, and that of dPCR may be as low as 0.01%. The turnaround time also favors dPCR over both sequencing technologies: 2 working days vs. 6 and 11 working days for Sanger sequencing and NGS, respectively. Cost per sample is around 200 EUR for Sanger sequencing, 90–100 EUR for NGS (when pooling eight samples on an Illumina MiSeq or Ion Torrent PGM), and from 3.5 to 11 EUR for dPCR.

The main advantages and disadvantages of Sanger sequencing, NGS, and dPCR for BCR-ABL1 KD mutation testing are shown in Table 2.

## 4. Conclusions and Future Perspectives

In CML, we currently stand at a turning point between old and novel methods for molecular testing. Indeed, in the hematology/oncology field, CML has been one of the first settings where novel technologies like dPCR and NGS have been explored for routine diagnostic use. This is mainly due to the fact that, over the past 20 years, sensitive molecular monitoring and mutation testing have become indispensable for optimal patient management. The development of different technologies in CML management and the number of PubMed citations reflecting their usage over the years are represented in Figure 4.

dPCR has proven to be a robust, sensitive, and accurate alternative to RT-qPCR in the monitoring of *BCR-ABL1* transcript levels, especially when DMR (a *sine qua non* condition to identify patients who are candidates for TFR) has to be assessed. As reported above [43], the promising results of the first international dPCR quality control round were recently presented, showing the better linearity from MR^1.0^ to MR^4.0^ of dPCR results as compared to RT-qPCR results. Similar observations were previously reported by an Italian collaborative group [45,72]. These might represent the first steps toward further larger cooperative efforts of harmonization of a quantitative PCR technique that is every day more attractive and promising in different clinical contexts. Regular international rounds of harmonization will indeed be necessary before dPCR might enter into the diagnostic routine worldwide.

As far as BCR-ABL1 KD mutation testing is concerned, over the past decade, NGS has proven to hold several advantages over Sanger sequencing in retrospective and prospective studies, such that the ELN 2020 recommendations have already incorporated the use of NGS for the assessment of patients not responding adequately to TKIs [2]. In this context, the use of dPCR to rapidly screen for some specific, highly relevant mutations (e.g., the T315I or the mutations known to confer resistance to the second-generation TKIs) is at its dawn, but holds great promise especially considering that dPCR is easier, faster, and more widely available than NGS. Future studies will be directed toward exploring the clinical advantage of the greater sensitivity offered by dPCR on one hand, and assessing how to best place dPCR and NGS testing (as alternative approaches, at least in some settings, or as complementary approaches) in the management of patients without an optimal response on the other hand.

The power of NGS might, in the future, be exploited well beyond the identification of TKI-resistant mutations in BCR-ABL1. Very recently, whole=exome sequencing and RNA-sequencing identified novel gene mutations and gene rearrangements or fusion transcripts in newly diagnosed CP CML patients, particularly those who ultimately experienced disease progression and poor outcome [73]. Indeed, the role of genomic and transcriptomic analyses in CML at the time of diagnosis, resistance, or progression has long been underestimated, but might prove important to empower patient management decisions, as already happens in other cancer conditions. Improved risk stratification based on scores integrating clinical and genomic features, translating into optimal selection of the most effective first-line TKI, is indeed an attractive opportunity toward which several groups are now working [74].

As technology evolves at a rapid pace, there are also several innovative approaches on the horizon. Some of them might be implemented in the diagnostic panorama of CML in the near future, which will be characterized by precise and personalized medicine [75]. Single-cell molecular approaches might prove useful to explore the cellular heterogeneity that might justify therapy resistance and disease progression. In CML, an interesting study using this technique demonstrated that CML leukemic stem cells isolated from TKI-resistant patients carry a transcriptional profile clearly distinct from that of normal hematopoietic cells and also from that of TKI-sensitive CML progenitors. This might offer scientists the possibility to design ab initio a personalized therapeutic project [76]. Lastly, quantum chemistry, which applies quantum mechanics to physical and chemical models assessing the ground state of atoms, is able to identify molecular changes induced by different treatments. Few data in this field are today available in the CML context, where this technique been focused especially on the TKI structure and on the interaction between TKIs and the ABL1 ATP-binding pocket. For example, vibrational spectroscopy clearly demonstrated that the beta form of imatinib was more stable than the alpha configuration, thereby helping to better understand the mechanism of action of this TKI [77]. Much work has to be done before these innovative approaches may prove their utility in CML. However, more ambitious treatment endpoints require a more sophisticated approach to stratify patients, monitor their response, and efficiently address drug resistance in a timely manner.

## Figures and Tables

**Figure 1 jcm-09-03865-f001:**
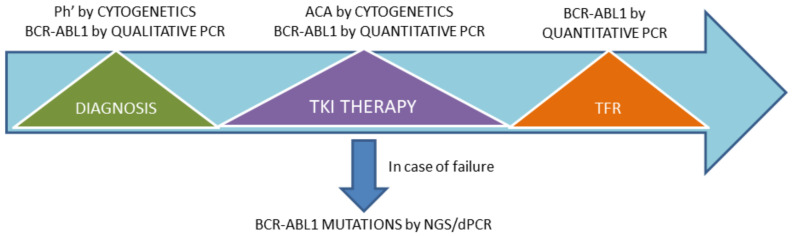
Flow chart of “biological” routine assessments to be performed in chronic myeloid leukemia (CML) patients according to the 2020 European Leukemia Net (ELN) recommendations. At diagnosis, it is mandatory to perform cytogenetics (chromosome banding analysis) to detect the Philadelphia (Ph1) chromosome and qualitative PCR to identify the *BCR-ABL1* fusion transcript type. During treatment, after the achievement of a complete cytogenetic response (no metaphases positive for the Ph1 chromosome out of a minimum of 20 metaphases examined), cytogenetic tests are advised only in cases with additional chromosomal aberrations (ACA) in the Philadelphia-positive clone, whereas quantitative PCR for *BCR-ABL1* has to be performed on a regular basis to monitor transcript levels (minimal residual disease (MRD)). Lastly, in treatment-free-remission (TFR), only quantitative PCR for *BCR-ABL1* transcript is needed. In cases of an unsatisfactory response, it is recommended to search for *BCR-ABL1* mutations by Sanger sequencing or next-generation sequencing (NGS).

**Figure 2 jcm-09-03865-f002:**
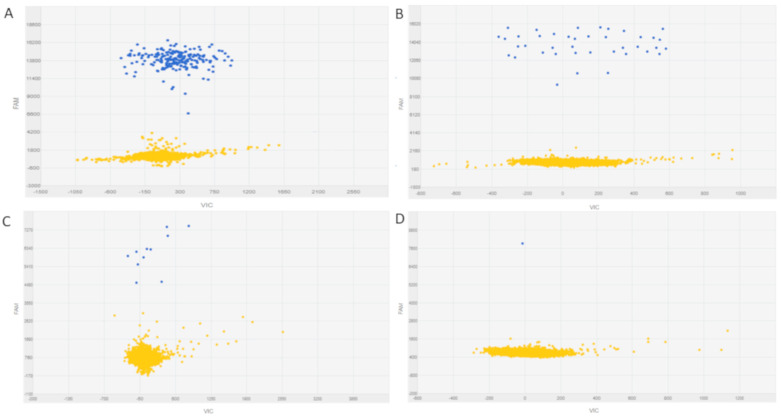
Examples of dilutions of BCR-ABL1 transcript detected by QuantStudio three-dimensional (3D) digital PCR platform. Blue dots represent the micro-reactions positive for BCR-ABL1. Yellow dots represent the micro-reaction negative for the presence of BCR-ABL1. (**A**) Detection of 200 copies of BCR-ABL1 transcript. (**B**) Detection of 50 copies of BCR-ABL1 transcript. (**C**) Detection of 10 copies of BCR-ABL1 transcript. (**D**) Detection of one copy of BCR-ABL1 transcript.

**Figure 3 jcm-09-03865-f003:**
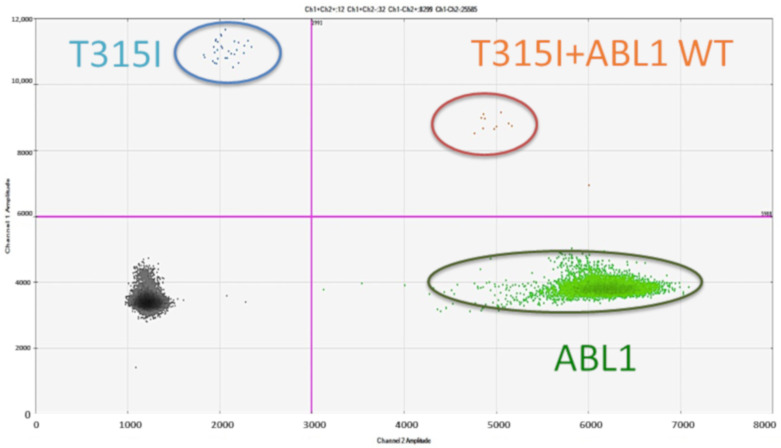
An example of application of dPCR for the detection of a mutation on genomic DNA. Here, a blue-labeled probe is designed to detect the T315I mutation and a green labeled probe is designed to detect unmutated *ABL1*. The quantitation of variant allele frequency (VAF) is obtained by the ratio between T315I and *ABL1* copies.

**Figure 4 jcm-09-03865-f004:**
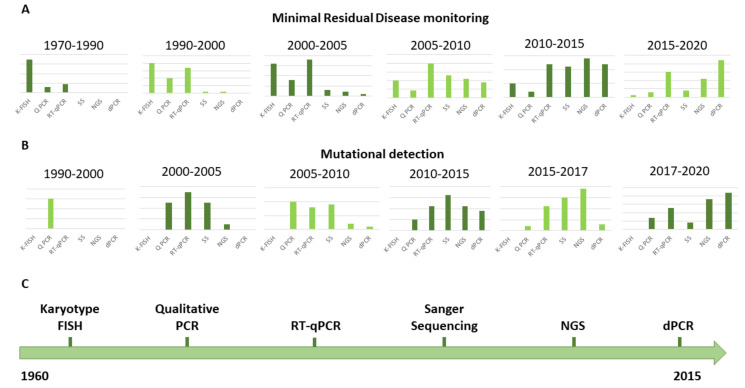
Evolution of the technologies for MRD and mutational status evaluation in CML patients. (**A**) The graphs report the number of publications referring to the application of different technologies for MRD monitoring in CML patients over the past 60 years. (**B**) The graphs report the number of publications referring to the application of different technologies for mutation detection in CML patients over the past 30 years. (**C**) The timeline reports the sequential introduction of different techniques for the MRD monitoring and the detection of mutations in CML patient management. Abbreviations: FISH: fluorescent in situ hybridization; K-FISH: karyotype analysis by chromosome banding and fluorescent in situ hybridization; Q PCR: qualitative polymerase chain reaction; RT-qPCR: real-time quantitative polymerase chain reaction; SS: Sanger sequencing; NGS: next-generation sequencing; dPCR: digital polymerase chain reaction.

**Table 1 jcm-09-03865-t001:** Summary of the main advantages and disadvantages of gold standard (RT-qPCR) and the innovative technology (digital PCR (dPCR)) for MRD monitoring in CML patients.

Method	Pros	Cons
RT-qPCR	Widely available Internationally standardized	Poor sensitivity and low precision at low levels of target Poor robustness Standard curve required
dPCR	More sensitive and accurate Enables the detection of as little as 1 copy of BCR-ABL1 transcript Performs an absolute quantification of the target without the need for a standard curve	Not yet widely available Not yet standardized

**Table 2 jcm-09-03865-t002:** Summary of the main advantages and disadvantages of old (Sanger sequencing) versus novel (NGS and dPCR) technologies for BCR-ABL1 KD mutation testing.

Method	Pros	Cons
Sanger sequencing	Widely available Easy to use	Poor sensitivity
NGS	More sensitive than Sanger sequencing Enables to scan the entire KD for any mutation Enables clonal analysis in case of multiple mutations falling within the same sequence reads (discrimination between compound and polyclonal mutations)	Not yet widely available Requires pooling of a minimum of 8–10 samples to be cost-effective Labor-intensive Not yet standardized RT-PCR and sequencing errors generate background “noise” at lower levels of sensitivity Chemistries and instruments still evolving
dPCR	Cheap, fast, and simple Has the greatest sensitivity	Can be implemented only for a limited number of mutations Not yet standardized May confirm the presence of compound mutations only if the mutation partners are already known and, hence, specific probes can be designed and used

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
