# Peer review of "Molecular Testing in CML between Old and New Methods: Are We at a Turning Point?"

_jcm, 2020, doi:10.3390/jcm9123865_

Round 1

Reviewer 1 Report

In general, the review is well-written and offers an up-to-date summary of the molecular techniques used in the BCR-ABL monitoring field.   I would request that the authors address a few minor points, as detailed below, and the review would benefit from proofreading for English language use.   Line 66: "which impacts on TKI sensitivity as well.(7) Clarify what is meant by this Line 73: limitations, such as? Line 255-310: the authors make a good comparison of dPCR and NGS but do not mention compound mutations. Can dPCR detect these? If not, it should be mentioned (albeit briefly) as an advantage of NGS. Line 284-287: How does the dPCR technique detect the mutations, as a multiplex? Why is dPCR more suitable for detecting mutations to second-generation TKIs while NGS better for imatinib-associated mutations? Please clarify as this is not clean. The review is missing a short phrase towards the end about other evolving technologies, e.g. RNA-seq, ultra-deep NGS, single cell techniques, quantum chemistry.

Author Response

We thank the Editors and the Reviewers for the constructive suggestions and comments, which really helped us to improve the manuscript. Please find below the point by point answer to the Reviewers’ questions.

REVIEWER 1

In general, the review is well-written and offers an up-to-date summary of the molecular techniques used in the BCR-ABL monitoring field. I would request that the authors address a few minor points, as detailed below, and the review would benefit from proofreading for English language use.

We have performed accurate proofreading of the review, and we have addressed the minor issues raised by this Reviewer as detailed below.

1) Line 66: "which impacts on TKI sensitivity as well. (7) Clarify what is meant by this

Thank you for this suggestion. We have modified the text to improve clarity. Please, see Lines 66-68

2) Line 73: limitations, such as?

We really appreciate this question and edited the text specifying the limitations we referred to. Please, see Line 78-79

3) Line 255-310: the authors make a good comparison of dPCR and NGS but do not mention compound mutations. Can dPCR detect these? If not, it should be mentioned (albeit briefly) as an advantage of NGS.

Thank you for this question. We agree with the Reviewer that this is a very important point that deserves to be addressed. For this reason, we i) modified Table 2 and ii) mentioned in the text how dPCR can be, in principle, implemented to detect compound mutations, citing an ASH abstract where this application was evaluated. Please, see Lines 337-346 and Table 2

4) Line 284-287: How does the dPCR technique detect the mutations, as a multiplex? Why is dPCR more suitable for detecting mutations to second-generation TKIs while NGS better for imatinib-associated mutations? Please clarify as this is not clean.

We have now modified the text in order to clarify how the multiplex dPCR can discriminate between mutations and why dPCR is more suitable to the detection of second-generation TKI-resistant mutations (that are associated to sixteen nucleotide substitutions) than to the detection of imatinib-resistant mutations (that are associated to more than sixty nucleotide substitutions). Please, see Lines 313-319

5) The review is missing a short phrase towards the end about other evolving technologies, e.g. RNA-seq, ultra-deep NGS, single cell techniques, quantum chemistry.

Thank you for this advice. We have completed the manuscript with a phrase concerning the evolving of new technologies in the field. Please, see Lines 413-439

We really thank the Reviewer for the accurate revision and the appreciated suggestions which really helped us in improving the quality of the manuscript.

Reviewer 2 Report

The manuscript by Soverini et al is fluent, well written and deals with two interesting topics related to chronic myeloid leukemia management: MRD molecular monitoring and BCR-ABL1 KD mutations. For each of the topics the main differences between old and new testing methods are described.

Only minor revisions are suggested:

  • Page 5, lines 165-169: please better clarify and give more details on the mentioned third generation dPCR chip-based platform
  • Page 5, lines 178-181: add more information about new commercial BCR-ABL1 digital assays
  • Page 5, lines 188-193: please introduce and explain MR4.7
  • Page 7, lines 255-261: explain more about the advantages and disadvantages of MinION technology for the identification of BCR-ABL1 mutations

Author Response

We thank the Editors and the Reviewers for the constructive suggestions and comments, which really helped us to improve the manuscript. Please find below the point by point answer to the Reviewers’ questions.

REVIEWER 2

The manuscript by Soverini et al is fluent, well written and deals with two interesting topics related to chronic myeloid leukemia management: MRD molecular monitoring and BCR-ABL1 KD mutations. For each of the topics the main differences between old and new testing methods are described.

Only minor revisions are suggested:

1) Page 5, lines 165-169: please better clarify and give more details on the mentioned third generation dPCR chip-based platform

Thank you for this suggestion. We have added more details concerning the new generation of dPCR platforms. Please, see Lines 94-98

2) Page 5, lines 178-181: add more information about new commercial BCR-ABL1 digital assays

We thank the reviewer for this advice. We have now included in the text information about the dPCR commercial assays for BCR-ABL1 detection. Please, see Lines 189-195

3) Page 5, lines 188-193: please introduce and explain MR4.7

Thank you for this point. The class MR4.7 is not comprised in the conventional MR classes belonging to the Deep Molecular Response classes. MR4.7 is referring a level of MRD between MR4.5 and MR5.0. In terms of percentage, it corresponds to 0.002 and it was reported also by Brown at al. We added this reference for clarity. Please see line 207

4) Page 7, lines 255-261: explain more about the advantages and disadvantages of MinION technology for the identification of BCR-ABL1 mutations

Thank you for this suggestion. We have now addressed the advantages and disadvantages of the MinION technology for BCR-ABL1 mutation detection. Please, see Lines 278-285

We really thank the Reviewer for the accurate revision and the appreciated suggestions which really helped us in improving the quality of the manuscript.

Reviewer 3 Report

In this manuscript, Soverini et al. discussed molecular testing in CML by reviewing the main features of old and novel technologies, providing an overview of the recently published studies assessing the potential clinical value of dPCR and NGS, and discussing how the state of the art might evolve in the next years. 

The manuscript includes some interesting new findings in research concerning the advancements in CML molecular testing. However, there are several comments that I have regarding the way the paper was written.

First, I think the paper requires extensive proofreading. For example: the sentences “Over the years… for TFR” (lines 49-51), “Indeed, the novel…many replicates” (lines 167-169), and “A gDNA… (FISH) (lines 138-141) exhibit different kinds of grammar mistakes.

Second, there are several pieces of information that are not cited in the papers, including “Overall, … (68% vs 43%)” (lines 102-105), “The cumulative…respectively” (lines 278-279), “All patients… relapse” (lines 279-281) and “Indeed... RT-qPCR” (line 190-192).

Third, some of the words and acronyms used require clarification, such as the word “actionable” (line 59), the undefined abbreviation of “VAFs” (line 304), and others which may leave the uninformed reader confused by what each carries for a meaning.

Furthermore, some of the authors’ conclusions are made without evidence of significant testing. In particular, the conclusion made in the paragraph starting at line 194 is based on the findings mentioned in the previous paragraph, which compares percentages of detection rates between two techniques but fail to provide confidence intervals and p values. In case significance testing was done in the cited source, then these should be mentioned to convince the reader about the conclusions made.

Author Response

We thank the Editors and the Reviewers for the constructive suggestions and comments, which really helped us to improve the manuscript. Please find below the point by point answer to the Reviewers’ questions.

REVIEWER 3

In this review Soverini et al. have tried to describe the old and novel techniques for molecular testing of CML. Although this review might somewhat add knowledge to what is already known in this field, it could be more interesting if the following points are considered for this:

1) An introductory table emphasizing on CML risk stratification based on validated cytogenetics and molecular abnormalities should be provided. This will help understand the basic concept for molecular testing.

We thank the Reviewer for the suggestion. According to the ELN guidelines, some additional chromosomal aberrations in the Philadelphia-positive clone can impair the outcome. On this basis, when these abnormalities are detected, we continue to follow patients with cytogenetics also after reaching complete cytogenetic remission. We have added a flow diagram about the schedule of cytogenetic and molecular analyses. Please, see Figure 1.

2) Authors should present a figure showing the timeline of development of the different testing technologies. Besides, statistics showing how well these tests have been used till date would be interesting

We really thank the Reviewer for this suggestion. We have added a figure showing the timeline of development of different technologies in MRD and mutation detection for CML patients, and a graph reporting the number of PubMed citations reflecting their usage over the years. Please, see Figure 4.

3) The importance of fluorescence in situ hybridization (FISH) in detecting the mutations must also be emphasized.

We thank the Reviewer for this suggestion. Despite its pivotal role in the diagnosis of CML, FISH is not recognized as a technique for point mutation detection in CML patients. According to international recommendations, the mutational status of the kinase domain is conventionally analyzed by PCR followed by Sanger Sequencing or by NGS. Very recently, dPCR have also been proposed. Unfortunately, we were not able to find any paper reporting the use of FISH for mutation detection in CML.

4) While comparing the various techniques authors should provide quantitative data for example, sensitivity (%), Turnaround time etc. 

We thank the Reviewer for this suggestion. We have now introduced in the text a comparison between the various techniques for mutation testing in terms of quantitative parameters like sensitivity, turnaround time and costs. Please, see lines 361-366.

5) How well have SNP arrays been used for CML testing must also be mentioned.

SNP arrays have indeed been used in the past in a few studies aimed to assess the frequency of copy number alterations in CML, especially in TKI-resistant cases (e.g., Nowak et al, Blood 2010; Boultwood et al, Leukemia 2010). However, they are not suitable either for MRD monitoring or for mutation detection purposes. Neither have they proven useful for prognostic stratification of patients, in contrast to acute lymphoblastic leukemia where deletion of IKZF1, PAX5, CDKN2A/B etc are associated with poorer prognosis. Thus, given that SNP arrays have never gained a prominent diagnostic role in CML, and considering that the theme of the review is focused on MRD monitoring and mutation detection, we respectfully believe that SNP arrays don’t find a place in the manuscript.

6) What is the authors take on the use of NGS for CML testing. Will it be affordable for patients? will the turnaround time delay the diagnosis of the patients? Do authors think using NGS for CML detection add an extra layer of confidence?

We thank the Reviewer for raising these issues. However, please note that NGS in CML will not be used with the purpose to confirm diagnosis of the patients. For that, cytogenetics and PCR are sufficient (following the suggestion of another Reviewer, we have added a Figure to clarify when the different tests are needed). The proposed role of NGS in CML is to screen patients who are not responding adequately to therapy for resistant mutations. In this setting, we have introduced a comparison of costs and turnaround times of the different methods. Please see lines 647-669

7) A comparative experiment showing the efficiency of detection between gold standard techniques and the new techniques would have been interesting.

We thank the Reviewer for this suggestion. In this review, many of the articles or meeting abstracts we refer to indeed report on studies aimed at comparative evaluation of new vs gold standard technologies, both for MRD monitoring and for mutation detection. We agree with the Reviewer that additional prospective trials will be needed to further confirm the value of dPCR, but this is out of the scope of the current manuscript.

Reviewer 4 Report

In this review Soverini et al. have tried to describe the old and novel techniques for molecular testing of CML. Although this review might somewhat add knowledge to what is already known in this field, it could be more interesting if the following points are considered for this :

1) An introductory table emphasizing on CML risk stratification based on validated cytogenetics and molecular abnormalities should be provided. This will help understand the basic concept for molecular testing.

2) Authors should present a figure showing the timeline of development of the different testing technologies. Besides, statistics showing how well these tests have been used till date would be interesting

3) The importance of fluorescence in situ hybridization (FISH) in detecting the mutations must also be emphasized.

4) While comparing the various techniques authors should provide quantitative data for example, sensitivity(%), Turnaround time etc. 

5) How well have SNP arrays been used for CML testing must also be mentioned.

6) What is the authors take on the use of NGS for CML testing. Will it be affordable for patients? will the turnaround time delay the diagnosis of the patients? Do authors think using NGS for CML detection add an extra layer of confidence?

7) A comparative experiment showing the efficiency of detection between gold standard techniques and the new techniques would have been interesting.

Author Response

We thank the Editors and the Reviewers for the constructive suggestions and comments, which really helped us to improve the manuscript. Please find below the point by point answer to the Reviewers’ questions.

REVIEWER 4

In this manuscript, Soverini et al. discussed molecular testing in CML by reviewing the main features of old and novel technologies, providing an overview of the recently published studies assessing the potential clinical value of dPCR and NGS, and discussing how the state of the art might evolve in the next years.

The manuscript includes some interesting new findings in research concerning the advancements in CML molecular testing. However, there are several comments that I have regarding the way the paper was written.

1) First, I think the paper requires extensive proofreading. For example: the sentences “Over the years... for TFR” (lines 49-51), “Indeed, the novel...many replicates” (lines 167-169), and “A gDNA... (FISH) (lines 138-141) exhibit different kinds of grammar mistakes.

We really thank the Reviewer for this point and for the accurate revision of the manuscript. We have proofread the paper and the text has been edited accordingly.

2) Second, there are several pieces of information that are not cited in the papers, including “Overall, ... (68% vs 43%)” (lines 102-105), “The cumulative...respectively” (lines 278-279), “All patients... relapse” (lines 279-281) and “Indeed... RT-qPCR” (line 190-192).

Thank you for this suggestion. We have improved the references and added citations all over the text.

3) Third, some of the words and acronyms used require clarification, such as the word “actionable” (line 59), the undefined abbreviation of “VAFs” (line 304), and others which may leave the uninformed reader confused by what each carries for a meaning.

We thank the Reviewer for this advice. We have clarified with a specific sentence what we meant for “actionable” and explained all the acronyms in the text. Please see Lines 59 and 304.

4) Furthermore, some of the authors’ conclusions are made without evidence of significant testing. In particular, the conclusion made in the paragraph starting at line 194 is based on the findings mentioned in the previous paragraph, which compares percentages of detection rates between two techniques but fail to provide confidence intervals and p values. In case significance testing was done in the cited source, then these should be mentioned to convince the reader about the conclusions made.

Thank you for raising this important point. We have added the p values and additional details reported in the cited paper in order to help the reader to understand our conclusions.

Round 2

Reviewer 3 Report

The authors did well in addressing my raised comments and concerns.

Reviewer 4 Report

Authors have responded to all the comments satisfactorily.